# Fast Spectral Approximation of Structured Graphs with Applications to Graph Filtering

**Mario Coutino** [1,*], **Sundeep Prabhakar Chepuri** [2], **Takanori Maehara** [3] **and Geert Leus** [1]

[1]   Department of Microelectronics, Faculty of Electrical Engineering, Mathematics and Computer Science, Delft University of Technology, 2628 CD Delft , The Netherlands; g.j.t.leus@tudelft.nl

[2]   Department of Electrical and Communication Engineering, Indian Institute of Science, Bangalore 560 012, India; spchepuri@iisc.ac.in

[3]   RIKEN Center for Advanced Intelligence Project, Tokyo 103-0027, Japan; takanori.maehara@riken.jp

[*]   Correspondence: m.a.coutinominguez@tudelft.nl

**Abstract:** To analyze and synthesize signals on networks or graphs, Fourier theory has been extended to irregular domains, leading to a so-called graph Fourier transform. Unfortunately, different from the traditional Fourier transform, each graph exhibits a different graph Fourier transform. Therefore to analyze the graph-frequency domain properties of a graph signal, the graph Fourier modes and graph frequencies must be computed for the graph under study. Although to find these graph frequencies and modes, a computationally expensive, or even prohibitive, eigendecomposition of the graph is required, there exist families of graphs that have properties that could be exploited for an approximate fast graph spectrum computation. In this work, we aim to identify these families and to provide a divide-and-conquer approach for computing an approximate spectral decomposition of the graph. Using the same decomposition, results on reducing the complexity of graph filtering are derived. These results provide an attempt to leverage the underlying topological properties of graphs in order to devise general computational models for graph signal processing.

**Keywords:** graph signal processing; graph Fourier transform; approximate graph Fourier transform; divide-and-conquer

## 1. Introduction

Graphs, as combinatorial objects, are important in many disciplines. They allow us to capture complex interactions between different entities (elements). Due to this ability, they have found applications in a plethora of fields, spanning from biology to psychology, and from logistics to medical sciences. Additionally, interest has emerged in adapting classical signal processing methods to deal with signals in irregular domains, e.g., signals defined on graphs [1,2]. As for traditional signal processing, the fundamental tool of graph signal processing is its analogous Fourier transform: the so-called *graph Fourier transform* (GFT). This transform interprets a signal on a graph through its graph *modes*. The concept of graph modes can be understood by considering the Laplacian matrix of a graph as a discrete representation of a manifold. The so-called graph modes are then the discrete counterpart of eigenfunctions of the Laplacian operator over the continuous manifold. Therefore, the graph modes can can be identified as the eigenvectors of the Laplacian matrix and can be obtained through its eigendecomposition. Here, we remark that although the analogy connecting with manifolds employs the Laplacian, the concept of GFT holds for different matrix representations of the graph such as the adjacency

matrix or normalized Laplacian [3]. For instance, let us consider an undirected (possibly weighted) graph $\mathcal{G} = (\mathcal{V}, \mathcal{E})$ defined on $n$ vertices with vertex set $\mathcal{V} = \{v_1, v_2, \ldots\}$ and edge set $\mathcal{E}$ and let $A$ and $D = \text{diag}(A\mathbf{1})$ be its (weighted) adjacency and degree matrices, respectively. Here, $[A]_{i,j} > 0$ only if $(v_i, v_j) \in \mathcal{E}$. Then, selecting a matrix representation of the graph, $S$, e.g., Laplacian, normalized Laplacian, adjacency matrix, we can define the graph Fourier transform of an $n$-dimensional signal $x$ over $\mathcal{G}$ as [3]

$$x_{\text{f}} = U^T x, \tag{1}$$

where $U$ is the orthogonal eigenvector matrix or the so-called *graph modes* obtained from the eigendecomposition of $S$, i.e., $S = U\Lambda U^T$, and $\Lambda$ is the diagonal matrix of eigenvalues, $\lambda = \{\lambda_1, \ldots, \lambda_n\}$ also known as *graph frequencies*.

Although there are several alternatives to select the matrix $S$, we follow the formalism of [4] and use the eigenvalues of the normalized Laplacian, i.e., $L_{\text{n}} := I - D^{-1/2}AD^{-1/2}$, to define the spectrum of $\mathcal{G}$ throughout this paper. Thus, the graph modes are then defined as the eigenvectors of $L_{\text{n}}$. We make this consideration here as the normalized graph Laplacian's eigenvectors define generalized translation and modulation operators, see, e.g., [3], and its spectrum relates well with graph invariants because its definition is consistent with those of eigenvalues in spectral geometry and stochastic processes [4], which is not the case for the spectrum of other choices of $S$.

As noted in [5], one of the main issues of performing the GFT is that a full eigendecomposition is required. This naturally incurs a $\mathcal{O}(n^3)$ cost, which for many applications is undesirable or even infeasible. Therefore, before aiming to achieve a complexity comparable to the fast Fourier transform (FFT), we require to obtain a better understanding of the eigendecomposition of $L_{\text{n}}$. In this context, a series of research works propose approximate fast GFT methods [5–7]. These methods employ Givens rotations and truncated Jacobi iterations to compute an orthogonal basis that approximately diagonalizes the matrix representation of the graph [8]. However, their construction relies on the assumption that Givens rotations, despite not being dense in the space of general orthogonal matrices, approximate well the matrix $U$. In addition, the main aim of these works is to obtain a fast GFT and not a fast graph spectral decomposition. That is, their focus is not on finding a basis for the transform efficiently, but to perform the action of the basis as fast as possible. Furthermore, no structure, as for example an excluded minor [9], is leveraged in those works. So, despite that these approaches address the problem, they fail to make use of particular structures (combinatorial in nature) that might improve the analysis and performance for restricted (but practical) families of graphs. Therefore, in this work, we aim to provide *a topological approach to devise a fast graph spectral decomposition*, i.e., a fast estimation of both $\Lambda$ and $U$, as a first step towards a fast GFT. Here, we argue that for some families of graphs there is a structure that can be leveraged to obtain divide-and-conquer algorithms for fast computation. In addition, we provide asymptotic theoretical bounds for the approximation of the spectrum of a graph up to any recursion level. Although the main aim of this work is to provide an approximate method for computing the spectrum and the modes of a graph, we provide theoretical bounds on the quality of this decomposition for other tasks such as graph filtering. These results open a venue, and describe a possible landscape, for further research on graph partitioning for graph filtering and hierarchical graph filtering.

## 2. Preliminaries

In this section, we introduce the notation that will be used throughout the rest of the paper as well as related works within the field of graph theory and graph signal processing (GSP). In addition, the main contributions of this work and the outline of the paper are presented.

## 2.1. Notation

We use calligraphic letters to denote sets with the exception of $\mathcal{G}$ and $\mathcal{H}$ that are reserved for representing graphs. We use $|\cdot|$ to denote the cardinality of a given set. $\deg(v)$ denotes the degree of vertex $v$. Upper (lower) case boldface letters such as $\boldsymbol{A}$ ($\boldsymbol{a}$) are used to denote matrices (vectors). $(\cdot)^T$ represents transposition. $\|\cdot\|$ represents a *generic* norm. The use of a particular norm will be clarified in the subscript. Throughout the paper, asymptotic notation is used. Therefore, when we use $\mathcal{O}(\cdot)$ and $o(\cdot)$ we always refer to big-O and little-o notation, respectively. They must be read as "is bounded above by" and "is dominated by", respectively. The notation $\tilde{O}(\cdot)$ is used to hide a factor of at most $(\log \log n)^4$. In addition, $\Omega(\cdot)$ denote big-Omega and it must be read as "is bounded below by". All these symbols describe the asymptotic behavior and hence represent the behavior in the asymptotic regime.

## 2.2. Prior Art

Besides the previously cited works that aim at finding a fast GFT, there are other works that closely relate to the existence of a fast scheme for computing a good graph spectrum approximation.

A close research area is the one of *graph sparsification*. Results in this area are widely used for solving large-scale linear systems involving Laplacian matrices [10]. In particular, graph sparsification and support theory [11] provide ways of finding optimal subgraph preconditioners for linear systems [12]. Another related area is the one devoted to *graph coarsening*. Here, instead of reducing the number of edges, reduced-size graphs are employed to approximate the action of a given graph [4]. In this context, a different interpretation of the well-known heavy-edge matching heuristic [13] for graph coarsening has been recently introduced in [14]. Similarly to our objective, the idea in [14] is to reduce the computational load of computing certain operations, e.g., quadratic forms, by introducing a trade-off between complexity and accuracy. So, while [14] aims to reduce the number of nodes involved in the computation of certain operations through stochastic arguments, here we aim to *break down* the graph in subgraphs to provide a trade-off in terms of accuracy and complexity but preserving the *whole spectrum* and not a subset of it. Furthermore, we give guarantees that are mostly *deterministic*, in contrast to the ones based on stochastic arguments in [14] leveraging the restricted isometric property [15].

Another related area is the one of graph signal downsampling [16]. Despite that this area is not directly connected to the goal of estimating the spectrum of a graph, specific combinatorial properties of the graph are exploited as in our work. For instance the symmetric property of the spectrum of bipartite graphs has recently been exploited to provide a theoretical framework for graph signal downsampling. Furthermore, using graph coloring ideas [9], this theory has been extended to arbitrary graphs with the aim to define wavelet filter banks [17]. Similar to these works, here we aim to take advantage of the combinatorial, algebraic and geometrical properties of certain families of graphs to provide useful simplifications of different graph signal processing tasks. One main difference between the bipartite-based approaches and our work is the fact that we focus on graphs which can be represented by (approximate) block diagonal matrices, whose off-diagonal blocks have *few* entries, e.g., $o(n)$, without requiring bipartite graph approximations. Our approach follows the spirit of those employed in domain decomposition methods for parallelization of linear system of equations ([18] Ch. 13.6), where graph partitions methods are used to distribute different part of the problem to the available processing units and later the solutions of the subproblems are aggregated to build the solution of the original problem.

Finally, the ideas presented in this work share similarities with decomposition methods used for different applications within signal processing and numerical linear algebra. For example, within GSP, multiscale (hierarchical) analysis modes have been proposed using recursive graph partitions based on the eigenvectors of the graph under study [19–21]. Similarly, clustering methods have been used for fast multiresolution matrix decomposition, see, e.g., [22–24]. In these works, the columns of the matrix to be

decomposed are collected in disjoint clusters, forming a block-diagonal matrix approximation, to speed up the computation of each decomposition level. However, while these methods provide powerful analysis tools, they do not address our problem of interest: graph spectral approximation.

## 2.3. Contributions

In this paper, we introduce results for the approximation of the spectrum of structured graphs. We focus on families of graphs that not only have good combinatorial properties allowing for a *good* and *efficient* approximation of their spectrum but are also relevant for practical applications such as spectral sums, e.g., log determinant, trace of inverse, or spectral histogram approximation. The major contributions of this work are the following:

- We provide an overview of families of graphs that accept good graph separators, i.e., graphs that can be separated into roughly equal subgraphs. Also, we discuss the characteristics of such families and how they relate to practical applications.
- Restricting ourselves to such graph families with good recursive graph separators, we propose a hierarchical decomposition for such families of graphs for approximate graph spectrum estimation.
- We propose a conquer mechanism to *stitch back together* the pieces of the hierarchical decomposition for approximate graph mode estimation.
- We provide a theoretical analysis, leveraging current results on graph spectrum similarity, for the proposed hierarchical decomposition. We derive asymptotic bounds for both the accuracy of the graph spectrum approximation and the computational complexity of the proposed method using the hierarchical decomposition. Through numerical simulations, we show that in practice the approximation of the graph spectrum, using such a decomposition, obeys the derived bounds.
- We employ our results to applications commonly found in the field of graph signal processing. In particular, we derive bounds for the accuracy of approximate graph filtering using the proposed decomposition for the considered families of graphs. Despite that this result shows that a straightforward application of decomposition does not provide an efficient approximation, it sheds light on the reach of such an approach. In addition, we show that the cumulative spectral density of a given graph, commonly employed for the design of graph filter banks, can be properly approximated using the proposed decomposition. This differs from other approaches used in practice which do not provide any approximation guarantees.

## 2.4. Paper Outline

The remainder of the paper is organized as follows. In Section 3 we introduce some necessary background of topological graph theory and graph separators. In Section 4, an approximation guarantee for the spectrum of a graph given its bisection is provided, while in Section 5 the divide-and-conquer algorithm and theoretical guarantees on its performance are described. Section 6 corroborates the performance of the proposed method through numerical experiments. Section 7 discusses the impact of the graph partitioning scheme on graph filtering and provides theoretical guarantees. Finally, Section 8 concludes the paper and indicates future research directions.

## 3. Topological Graph Theory

Topological graph theory is a field within mathematics that studies the spatial embeddings of graphs, as it considers graphs as topological spaces [25]. From a computational perspective, results in algorithmic topological graph theory combine methods from computational geometry and data structures with classical methods from combinatorics, algebraic topology, and geometry. The recent advances in this area have

led to algorithms applicable in many different areas of computer science. As results within this area are extensive, and several of them are outside the scope of this work, we refer the interested reader to [26] and references therein for an overview of this field.

In this work, we mostly focus on results from *graph separators* for particular families of graphs. Such graph separators provide partitions of graphs with desirable properties in terms of size and graph topology. Before presenting how to use these partitions to devise algorithms for estimating the spectra of a graph, we present a brief introduction to graph separators.

### 3.1. Graph Separators

Let us consider an undirected graph $\mathcal{G} = (\mathcal{V}, \mathcal{E})$, where $\mathcal{V}$ and $\mathcal{E}$ denotes its vertex set and edge set, respectively. In many instances, we desire to find a partition $\{\mathcal{P}_i\}_{i=1}^Q$ of the vertices in $\mathcal{G}$, i.e., $\mathcal{V} = \cup_{i=1}^Q \mathcal{P}_i$, such that the members of such groups are balanced, i.e., $|\mathcal{P}_i| \approx |\mathcal{P}_j|$, $\forall\, i, j \in \{1, 2, \ldots, Q\}$, and either (i) the number of edges connecting the different partitions is minimized or (ii) the number of vertices that are needed to be removed to disconnect the graph is minimized. A common partition strategy is based on the field of *graph separators* [27].

In general, there are two kinds of graph separators: *vertex separators* and *edge separators*. In the case of vertex separators (see Figure 1), the vertices $\mathcal{V}$ are partitioned in three sets $\mathcal{A}$, $\mathcal{B}$ and $\mathcal{C}$. The goal of this partition is to disconnect $\mathcal{A}$ and $\mathcal{B}$ while minimizing the number of vertices in $\mathcal{C}$ and maintaining a balance between $\mathcal{A}$ and $\mathcal{B}$, i.e., $|\mathcal{A}| \approx |\mathcal{B}|$. Since there are no edges between $\mathcal{A}$ and $\mathcal{B}$, vertex separators are also known as *bisectors* or *bifurcators*.

For the case of *edge separators* (see Figure 2), the vertices are partitioned in two balanced sets, $\mathcal{A}$, and $\mathcal{B}$ such that the number of *cut edges*, i.e., edges between the two sets, is minimized.

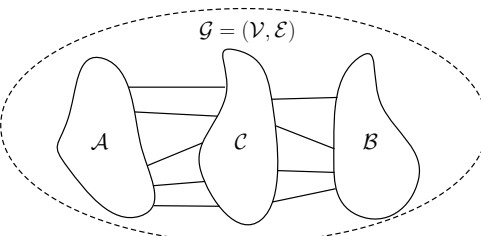

**Figure 1.** Illustration of a vertex separator. The vertex set $\mathcal{V}$ is divided in three subsets $\mathcal{A}$, $\mathcal{B}$, $\mathcal{C}$ such that $\mathcal{A}$ and $\mathcal{B}$ are disconnected, while minimizing the size of $\mathcal{C}$ and maintaining a balance (in terms of the number of vertices) between $\mathcal{A}$ and $\mathcal{B}$.

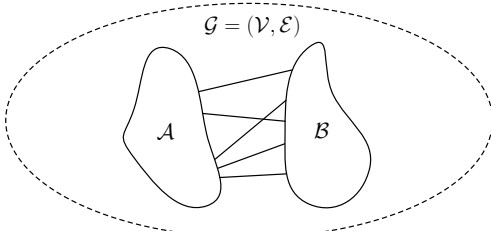

**Figure 2.** Illustration of an edge separator. The vertex set $\mathcal{V}$ is divided in balanced subsets $\mathcal{A}$ and $\mathcal{B}$ such that the number of edges between them are few.

As there are (possibly) many ways to *separate* a graph, we first make the following remarks:

(i)   Finding optimal graph separators is *hard*.

(ii)  Graphs whose graph separators are *sublinear* in size are considered as *good graph separators*. That is, the number of vertices (edges) that needs to be removed to partition the graph in balanced sets is $o(n)$.

(iii)    Graphs with *good recursive* graph separators are graphs whose resulting partitions have *good separators* themselves. For instance, if the class $\mathcal{S}$ of graphs has good graph separators, and $\mathcal{S}$ is closed under vertex (edge) deletion, then the graphs in $\mathcal{S}$ have good recursive graph separators.

Now, we clarify further these remarks. We make the distinction between graph separators and *good graph separators* due to the fact that we are interested in graphs that can be partitioned in balanced sets by removing just a *few* vertices (edges). Since algorithmic reasons dictate properties that can be propagated to the subsequent partitions, we stress the importance of graphs with *good recursive separators*. This is important for the design of divide-and-conquer algorithms or for parallel processing because it implies the possibility of generating increasingly small graphs by recursively splitting the problem into many subproblems. Therefore, identifying structured graphs accepting good recursive separators are of high importance from an algorithmic perspective. Finally, despite the discouraging result (i), fortunately, there are types of graphs that have good separators [cf. (ii)] and there exist efficient algorithms for finding them.

As a last comment, let us discuss briefly the relationship between vertex and edge separators. An important result, linking vertex and edge separators, is that a good edge separator for a graph $\mathcal{G}$ automatically provides a good vertex separator for $\mathcal{G}$ given that the restriction on the balance of the sets is not very strict. Although the converse of this statement is not true in general, for bounded degree graphs (as is the case for most graphs encountered in real applications) it can be shown that the definition of an edge separator and vertex separator is equivalent up to a constant. Therefore, good vertex separators can be obtained from good edge separators (and vice versa) for such types of graphs. As in this work we are interested in partitioning a graph into disjoint sets, we mostly focus on edge separators. However, unless it is explicitly stated, throughout this work when we refer to graph separators we do not further make the distinction between edge or vertex separators.

*3.2. Structured Graphs with Good Graph Separators*

Let us introduce the following definition by Lipton and Tarjan [28].

**Definition 1.** [$(\alpha, \beta)$-separator] *A class $\mathcal{S}$ of graphs satisfies the $f(n)$-separator theorem for constants $\alpha < 1$ and $\beta > 0$ if every n-vertex graph in $\mathcal{S}$ has a vertex partition $\mathcal{A} \cup \mathcal{B} \cup \mathcal{C}$ such that*

$$|\mathcal{A}|, |\mathcal{B}| \leq \alpha n, \tag{2a}$$
$$|\mathcal{C}| \leq \beta f(n), \tag{2b}$$

*and no edge has one endpoint in $\mathcal{A}$ and the other in $\mathcal{B}$. Here, $f(n)$ denotes a function, e.g., polynomial, of n.*

From the definition of $(\alpha, \beta)-$separator, families of graphs with $\alpha \approx 1/2$, $f(n) \in o(n)$ and that are closed under bisection, i.e., the induced graphs by the subsets are contained in the family itself, are the main candidates for families of graphs with good recursive separators.

Although many sparse graphs do not have a nontrivial separator theorem in terms of Definition 1, there are many families that do. Examples of families that have good separators and are closed under bisection are planar graphs [28], e.g., graphs related to meshes and computer graphics, finite element graphs [28], e.g., graphs arising from tesselation of the space, and geometric graphs [29], e.g., *k*-nearest-neighbor graphs, sensor networks. Fortunately, these graph families enclose graphs that we encounter in typical (graph) signal processing applications such as field estimation, distributed sensor networks, smart grids, to name a few. For example, in the network of connections of a person, the sets depicted in Figure 1 could represent their family and close friends ($\mathcal{A}$), their coworkers or professional

network ($\mathcal{B}$) and their acquaintances which interface with both networks ($\mathcal{C}$). In this setting, it is sensible to assume that the number of people that interface with both subnetworks is *small* as in most cases these two domains barely intersect. Therefore, we can be encouraged to think that, similar to the fast Fourier transform (FFT), for graphs within these families we can devise algorithms for problems in GSP leveraging graph separators. For more details on these and other graph families, the reader is referred to Appendix A where more examples are the different families are explained in detail.

## 4. Approximate Graph Spectrum

Now that we know there exist graphs with good (recursive) separators, what is left to answer is the following: *how well can we approximate the spectrum of a graph given a graph separator?*

To address this question, we introduce our first result.

**Theorem 1.** *Let $\mathcal{G} = (\mathcal{V}, \mathcal{E})$ be a graph for which a $(\alpha, \beta)$-separator exists. Further, consider $\mathcal{H} = (\mathcal{V}, \mathcal{E}')$ as the graph induced by removing the edges that connect the partition elements of $\mathcal{G}$ obtained by the $(\alpha, \beta)$-separator. If $\max_{v \in \mathcal{V}} \deg(v) \in \mathcal{O}(1)$, then*

$$W_1(\lambda, \lambda') \in \mathcal{O}\left(\frac{f(n)}{n}\right), \tag{3}$$

*where $\lambda$ and $\lambda'$ are the spectra of $\mathcal{G}$ and $\mathcal{H}$, respectively, and $W_1(\lambda, \lambda')$ is the Wasserstein distance between the discrete spectra $\lambda$ and $\lambda'$.*

**Proof.** See Appendix B.　□

This result motivates the use of graph separators for graph spectrum approximation. It shows that for families of graphs with *good separators*, i.e., $f(n) \in o(n)$ and a maximum degree independent of the size of the graph, e.g., in a road map the number of highways connecting cities does not increase with the number of cities, we can obtain a good approximation of the spectrum of $\mathcal{G}$ through the spectrum of $\mathcal{H}$. The main advantage of computing the spectrum of $\mathcal{H}$ instead of the spectrum of $\mathcal{G}$ is that the spectrum of $\mathcal{H}$ is disjoint. That is, its spectrum is the union of the spectra of its components, i.e., the matrix representation of $\mathcal{H}$ is a block diagonal matrix (after node permutation). This leads to a divide-and-conquer approach for approximating the spectrum of $\mathcal{G}$ through the one of $\mathcal{H}$. By computing the spectrum of the reduced-size graphs induced by the partition, and stitching the result afterwards (conquer), an approximation of the spectrum of $\mathcal{G}$ can be obtained. In the following, we exploit this idea to provide an algorithm, as well as theoretical guarantees, for approximate fast graph spectrum computation of families of graphs that accept *good graph separators*. Furthermore, we also extend this method for families of graphs with *good recursive separators* which accept a hierarchical decomposition.

## 5. Divide-And-Conquer for Fast Graph Spectrum Estimation

Conceptually, the proposed divide-and-conquer approach is straightforward. Starting from a large graph, we perform, recursively, bisections, i.e., partition the graph in two subgraphs, until the subgraphs are of a *manageable size*, i.e., computations are affordable. Then, the spectra of the reduced-size graphs are obtained and later *stitched* bottom-up until the top level is reached. This approach is analogous to the divide-and-conquer approach used for computing the spectrum of tridiagonal matrices [30]. In this particular case, it is easy to see that the "graph" can be split directly in the "middle", i.e., in the middle of the rows and columns. The two parts are *approximately* disjoint, hence after computing the spectrum of each part the two parts can be stitched to obtain an exact spectrum. Notice, that although we use the term "exact", a more appropriate term is "$\epsilon$-close" spectrum. The divide-and-conquer (DC) method is summarized in Algorithm 1 and an illustration of the approach is shown in Figure 3. In this figure, it is

shown how a binary tree is generated by recursive bisections until the desired depth is reached (divide). Afterwards, the spectrum is constructed at each parent node, in a bottom-up fashion, by the union of the spectra of its children (conquer).

---

**Algorithm 1:** `DC_Approx_Graph_Spectrum` $(\mathcal{G}, h, d)$.

**Input:** Graph $\mathcal{G}$, function for computing eigenvalues $h : \mathcal{S} \mapsto \lambda$ for graph family $\mathcal{S}$, and depth $d$.

1 **if** $d == 0$ **then**
2 $\quad \hat{\lambda} \leftarrow h(\mathcal{G})$
3 **else**
4 $\quad \{\mathcal{H}_1, \mathcal{H}_2\} \leftarrow$ `bisectGraph`$(\mathcal{G})$;
5 $\quad \lambda_1 \leftarrow$ `DC_Approx_Graph_Spectrum`$(\mathcal{H}_1, h, d-1)$;
6 $\quad \lambda_2 \leftarrow$ `DC_Approx_Graph_Spectrum`$(\mathcal{H}_2, h, d-1)$;
7 $\quad \hat{\lambda} \leftarrow$ `merge`$(\lambda_1, \lambda_2)$;
8 **end**

**Result:** $\hat{\lambda}$ : Approximate spectrum of $\mathcal{G}$

---

Notice that in Algorithm 1 there are several things left undefined: (i) the function $h$ which maps a graph to its spectrum, e.g., standard eigenvalue decomposition, (ii) the depth of the binary tree and the (iii) merge routine, e.g., union of elements. Although in principle, the method can be applied for any given functions $h$, `merge`, and depth $d$, we decided to keep the algorithm description as general as possible due to the fact that these three free parameters impact the method in the following ways.

(a) The function $h$ directly affects the computational complexity as it is the core operation of the algorithm. In addition, if $h$ is not an exact algorithm but a randomized or approximate method, then it will impact as well the quality of the final approximation.

(b) The `merge` function is the *conquer* step that stitches back the solution of the leaves. As this function is called at every non-leaf node, dealing with increasing-size arguments (traversing the tree upward), it must be a low-complexity routine to not blow up the computational complexity of the method.

(c) The depth of the tree, i.e., the number of recursive bisections, affects both the computational complexity and approximation quality. As each bisection worsens the approximation, deeper binary trees (theoretically) worsen the quality of the approximated spectrum. In addition, as the depth parameter controls the base case of the recursion for applying $h$, its value also impacts the final complexity of the algorithm.

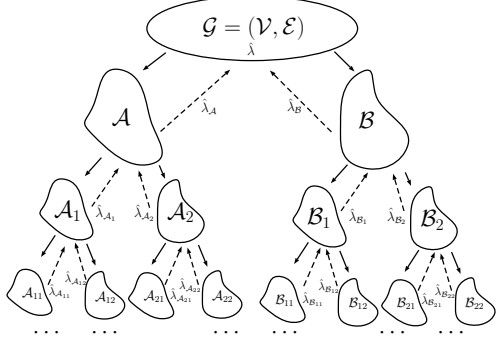

**Figure 3.** Illustration of the divide and conquer approach described in Algorithm 1. First, a binary tree with successive graph bisections is constructed (divide). The graph spectrum is reconstructed by stitching back together the pieces of the spectra of the reduced-size graphs (conquer).

An interesting property of this method is that it can be extended to other kinds of tasks. That is, this divide-and-conquer method can be used in all applications involving spectral properties of large graphs. For instance, we could consider applications where we desire to test if there exists an eigenvalue within a certain interval in a very large graph. By following this approach, if the graph under test falls within one of the previous families, we can test this property on the reduced-size graphs. However, there is always a trade-off between approximation quality and complexity, i.e., higher bisection orders worsen the performance derived from Theorem 1.

### 5.1. Accuracy-Complexity Trade-Off

Until this point, we only have provided theoretical guarantees for a graph decomposition with a single level, i.e., depth $d = 1$. In the following, we introduce the main theorems of this work with respect to both the quality and complexity of the DC approach for graph spectrum approximation. For this analysis, we consider the case where the `merge` routine solely sorts its input, incurring a complexity of $O(n \log n)$ for inputs of size $n/2$ each.

Before proceeding, we first introduce the following definition.

**Definition 2.** [Linearly separable graph family] *The set $\mathcal{S}_c$ of graphs for which a $\mathcal{O}(n^c)$-separator theorem exists, with $c < 1$, and whose subgraphs are also contained in the set, is called a c-separable graph family. Furthermore, if the separator is efficiently computable in (approximately) linear time then the set is a c-linearly separable graph family and it is denoted as $\mathcal{S}_c^l$.*

For the sake of simplicity, we restrict our results to the family of graphs $\mathcal{S}_{1/2}^l$. This family includes planar graphs, geometric graphs in 2-D, and graphs with a restricted minor [9]. However, the results hold for any *c*-separable graph family as long as the exponents are modified appropriately. In the following, the main results with respect to approximate graph spectrum estimation are presented.

**Theorem 2.** *For a graph $\mathcal{G} \in \mathcal{S}_{1/2}^l$, Algorithm 1 with a decomposition tree of depth given by $d^* = \mathcal{O}((1 - 2\delta) \log n)$ for $\delta \in (0, 1/2)$, provides an $\mathcal{O}(1/n^\delta)$-approximation of the spectrum of $\mathcal{G}$ w.r.t. the Wasserstein distance in $\mathcal{O}(n^{1+4\delta})$ time.*

**Proof.** See Appendix C. □

From this result, two direct corollaries providing the limits of the proposed method for the family $\mathcal{S}_{1/2}^l$, in terms of accuracy and time complexity, can be directly obtained as follows.

**Corollary 1.** *An $\mathcal{O}(1)$-approximation (constant factor) for the spectrum of $\mathcal{G} \in \mathcal{S}_{1/2}^l$ is achievable in $\mathcal{O}(n \log n)$ time when the depth of the decomposition tree for Algorithm 1 is chosen to be $d = \lceil \log n \rceil$.*

**Corollary 2.** *An $\mathcal{O}(1/\sqrt[4]{n})$-approximation for the spectrum of $\mathcal{G} \in \mathcal{S}_{1/2}^l$ is achievable in $\mathcal{O}(n^2)$ time when the depth of the decomposition tree for Algorithm 1 is chosen to be $d = \lceil \frac{1}{2} \log n \rceil$.*

These results provide the following two insights. First, for large-size graphs the approximation is tight, i.e., it vanishes as *n* grows. Hence, the proposed method is *asymptotically efficient* in the size of the graph. Second, the time complexity of the method is bounded below by the so-desired $\mathcal{O}(n \log n)$ complexity, and above by the (naive) cubic complexity, $\mathcal{O}(n^3)$. Therefore, as pointed out in Corollary 2, a good compromise between the decay rate of the error and complexity can be achieved when we fix $\delta$ to 1/4.

Although a discussion about parallelization is not the main aim of this work, we want to stress that, as most of the divide-and-conquer approaches, the proposed method is *parallel friendly*. That is, as each

of the subproblems (at the leaves) are disjoint, they can easily be parallelized providing a significant reduction in the time complexity. However, this analysis is outside the scope of this work and it does not provide any further insights into the proposed method.

*5.2. Eigenvectors Computation*

Although we have mainly focused on the computation of the spectrum of the graph, for the sake of completeness, in the following, we present some theoretical results that provide insights on the approximation quality of the graph eigenvectors. These results motivate the later discussion of the landscape for the approximation of graph filters in Section 7.

To bound the error of the eigenvectors obtained through the proposed decomposition, we make use of the Bauer-Fike theorem [31] and provide the following result.

**Theorem 3.** *Let $S$ be the matrix representation of graph $\mathcal{G}$. Further consider $\hat{S}$ as the matrix representation of the graph $\mathcal{H}$ obtained from $\mathcal{G}$ at depth $d$. If $\hat{u}$ is the eigenvector of $\hat{S}$ with eigenvalue $\hat{\mu}$, then there exists an eigenvector $u$ of $S$, with eigenvalue $\mu$, such that*

$$\frac{1}{n}\|\hat{u} - u\|_2^2 \leq \frac{4\gamma}{\min_{l \in \mathrm{eig}(S), l \neq \mu}|l - \mu|^2}. \tag{4}$$

*where $\gamma := \frac{1}{n}\|S - \hat{S}\|_F^2$.*

**Proof.** See Appendix D. □

Despite that the above result shows a vanishing behavior in the normalized Euclidean distance between the approximated and true eigenvectors if $\gamma \in o(n)$, it also shows that for highly clustered eigenvalues, the approximation error can be large. This result is not surprising as all eigendecomposition methods present difficulties for defining the eigenspace of highly clustered eigenvalues.

A possible way to improve the approximation error for the eigenvectors is through iterative refinement. Here, we will use the inverse iteration method to show that it is possible to perform improvements in near-linear time. The result is presented in the following theorem.

**Theorem 4.** *Let $(\hat{\mu}, \hat{u})$ be an $\mathcal{O}(1/n^\delta)$-approximated eigenpair of the matrix representation, $S$, of a graph $\mathcal{G} = (\mathcal{V}, \mathcal{E})$ with $m$ edges. Then, the kth iteration of the inverse iteration method, i.e.,*

$$\hat{u}_{k+1} = \frac{1}{c_k}(S - \hat{\mu}I)^{-1}\hat{u}_k, \tag{5}$$

*where the normalization constant $c_k$ is appropriately chosen, produces an eigenvector estimate $\hat{u}_{k+1}$ such that*

$$\|u - \hat{u}_{k+1}\|_2 \in \mathcal{O}\left(\left|\frac{1/n^\delta}{\mu - \mu_{\mathrm{close}}}\right|^k\right), \tag{6}$$

*where $\mu$ is the closest eigenvalue of $S$ to $\hat{\mu}$ and $\mu_{\mathrm{close}}$ is the closest eigenvalue of $S$ to $\lambda$. Moreover, such an $\hat{u}_{k+1}$ can be computed in expected time $\tilde{\mathcal{O}}(m\log^2 n \log 1/\eta)$ at $\eta$-accuracy when $S$ is the combinatorial Laplacian. For cases where $S$ is the graph adjacency matrix, the complexity result holds for eigenpairs with $\hat{\mu} \geq \max_{v \in \mathcal{V}} \deg(v)$.*

**Proof.** See Appendix E. □

Although the proposed improvement of the approximation quality of the eigenvectors is based on an iterative method, the inverse iteration method tends to convergence in few iterations (typically one) when the approximate eigenvalue is close to the true one. Finally, we stress that the previous result is not the tightest in terms of computational complexity. If further structure of the Laplacian is known, e.g., planarity, faster solvers [32] can be employed to speed up the computation of the inverse iteration method. Notice that although the power method could have been used to compute the modes of the graph, having available $\mathcal{O}(1/n^\delta)$-approximations for all eigenvalues leads to an overall reduction of the computational effort.

## 6. Experimental Results

To illustrate the performance of the proposed decomposition, in the following we perform a set of numerical experiments. In these numerical examples, we mostly focus on the family $\mathcal{S}_{1/2}^l$ as it includes most of the commonly encountered graphs within the field of graph signal processing, e.g., meshes, sensor networks, etc. For these experiments, we make use of the graph separator toolbox [33].

### 6.1. Depth of Hierarchical Decomposition

To validate the theory, we first perform numerical simulations for graphs in $\mathcal{S}_{1/2}^l$ of fixed dimension for different decomposition depths. Here, we generate a set of 100 random $\mathcal{S}_{1/2}^l$ graphs with $n = 1000$ nodes. The model to generate such graphs follows a unit disk model (see Appendix A). The model to generate such graphs follows a unit disk model (see Appendix A). Here, we generate $n$ 2D points in $[0,1] \times [0,1]$ and establish edges between points pairs whose distance is below a fixed threshold. For this simulation, we have set the threshold to 0.1.

From Figure 4a we can observe how the average of the normalized approximation error, i.e., $e_n = \|\lambda - \hat{\lambda}\|_2 / \|\lambda\|_2$ follows the expected behavior from the developed theory. That is, there is a rapid increase of the approximation error for higher decomposition orders, i.e., increasing the depth parameter. From this plot, it can be observed that when the depth parameter is fixed to $d^* = \lfloor 0.5 \log_2(n) \rfloor = 4$, the theoretical normalized approximation error is below 20% while the empirical error is below 10%. This discrepancy is due to the worst-case analysis for the derived error and the omitted additive and multiplicative constant hidden by the asymptotic notation. This is the depth at which the method provides a good trade-off between computational complexity and spectrum approximation accuracy as described in the theoretical analysis.

### 6.2. Asymptotics for Graph Size

Similarly, we perform an analysis to evaluate the behavior of the proposed method in terms of the graph size. Here, we set a fixed depth of $d = \{4, 5, 6\}$ and perform a series of experiments for graphs of different sizes within the family $\mathcal{S}_{1/2}^l$. For each graph size, we generate 100 random graphs and the average normalized approximation error, $e_n$, is reported in Figure 4b.

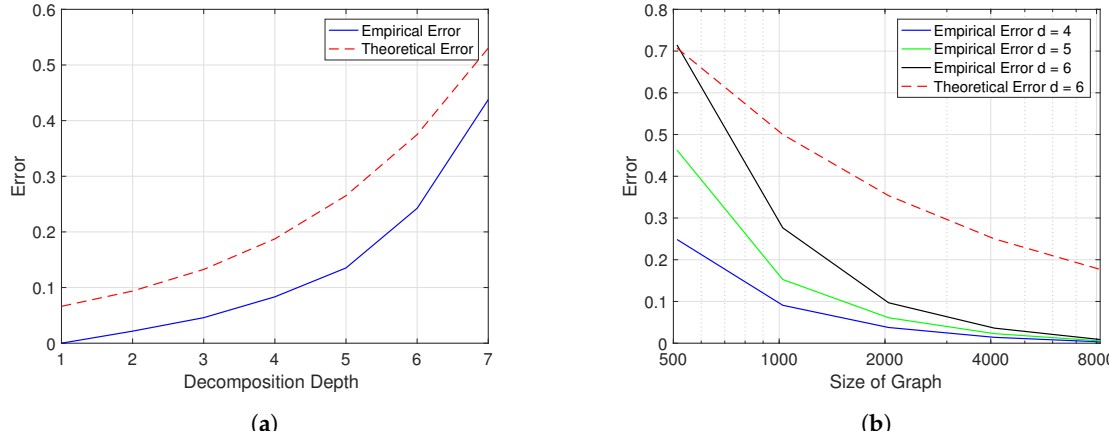

(**a**)                                         (**b**)

**Figure 4.** (**a**) Approximation quality of the graph spectrum for the family $\mathcal{S}^{l}_{1/2}$, with $N = 1000$, for different depths of the hierarchical decomposition. (**b**) Approximation quality of the graph spectrum for the family $\mathcal{S}^{l}_{1/2}$, with decomposition depth $d = \{4, 5, 6\}$, for graphs of different sizes.

As before, the trend of the normalized error follows the expected behavior provided by the theoretical analysis. That is, for graphs with increasing number of nodes, the quality of the approximation improves. This result illustrates the asymptotically efficient behavior of the proposed method for spectrum estimation using the hierarchical decomposition for these families of graphs.

*6.3. Time Complexity*

At this point, we have shown that the theory holds for the approximation quality of the graph spectrum. However, one of the appealing reasons to use this approach is the possibility of making a trade-off between spectrum accuracy and computational complexity. To show the behavior of the proposed method, in Figure 5 we plot the time required by the built-in MATLAB function $eig(\cdot)$ for different graph sizes and compare it with the time required to obtain the approximate spectrum with the method introduced in this work.

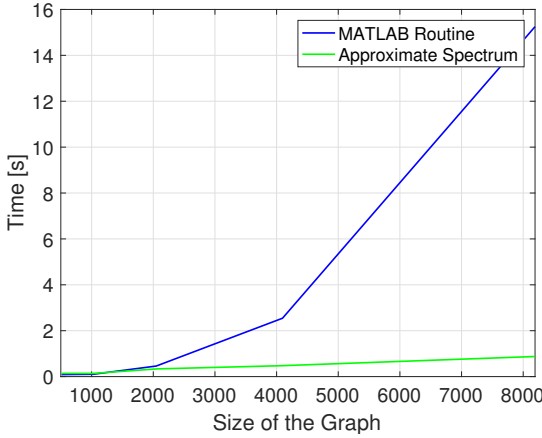

**Figure 5.** Comparison of the time required by the built-in MATLAB routine for obtaining the eigenvalues of a matrix, and the time required by the proposed algorithm levering a hierarchical decomposition.

Clearly, the time required by the MATLAB function follows the expected complexity for eigenvalue decomposition. However, the proposed approximation method does not exhibit such a rapid increase for the different tested graph sizes. For this simulation, the depth decomposition is set to $d = 0.5 \log_2(n)$ as this is the depth value that provides a good trade-off between computational complexity and approximation accuracy. The time reported for the proposed method includes (i) the time for the decomposition, (ii) the time for computing the eigenvalue decomposition at the leaves and (iii) the time for sorting the eigenvalues. The combination of these results shows that both in theory and practice the proposed method delivers good and feasible approximations for the spectrum of structured graphs. This opens a venue for approximate signal processing techniques that leverage the underlying combinatorial structure of the graph.

*6.4. Real Example: Minnesota, a Close-To-Planar Graph*

To further illustrate the applicability of the proposed method for another choices of $S$, we use a well-known graph, the Minnesota graph [34], and the combinatorial Laplacian to show that this method can be applied to real graphs that approximately meet the conditions discussed in previous sections. It is known that the Minnesota graph is not a planar graph as it has a subgraph isomorphic to $K_{3,3}$, the complete balanced bipartite graph on six vertices [35]. However, through this example, we show that it is possible to obtain an approximate spectrum, and its cumulative spectral density. As the Minnesota graph has $n = 2642$ nodes, we set a decomposition depth of $d = 5$. Using the geographic coordinates (as each node is located in a 2D space) we perform the hierarchical decomposition by recursively divide the graph using geometric partition [29]. This partition provides us with a good separator at every depth and its computation requires linear effort, i.e., $\mathcal{O}(2n)$. From the approximate spectrum, we compute the cumulative spectral density [36] using the expression

$$P_\lambda(z) = \frac{1}{N} \sum_{l=0}^{N-1} \mathbb{1}_{\{\lambda_l \leq z\}}, \tag{7}$$

where $\mathbb{1}_{\{\lambda_l \leq z\}}$ denotes the indicator function which takes a value of one whenever $\lambda_l \leq 0$ or zero otherwise.

The comparison between the true spectrum and cumulative spectral density with their approximations is shown in Figure 6. In addition, in Figure 6a we have color-coded the different subsets of nodes that belong to each of the elements of the partition. For this particular graph, the built-in MATLAB function $eig(\cdot)$ takes approximately $\approx 9$ s for computing the matrix eigenvalues, while the method based on the hierarchical decomposition only takes $\approx 0.3$ s. Finally, we show in Figure 7 some measures related to the quality of the approximated eigenvectors, $\hat{U}$, of the graph. Particularly, in Figure 7a, we show the absolute value of the inner product between the approximate and true graph eigenvectors. Here, we observe a banded structure whose bandwidth decreases for eigenvectors related to larger eigenvalues. Simiarly, in Figure 7b, we observe that the approximate eigenbasis almost diagonalizes the original graph Laplacian. For completeness, we show in Figure 7c a comparison in terms of the structure, i.e., Laplacian matrix support, of the original graph (red and blue dots) after permutation based on the partitioning and the block-diagonal approximation (blue dots).

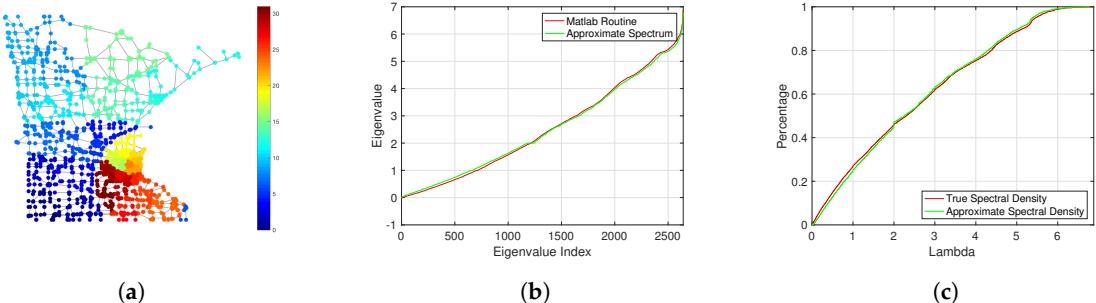

**Figure 6.** Illustration of the method for the Minnesota graph, a close-to-planar graph. (**a**) Comparison of the true (red) and approximate (green) spectrum of its Laplacian matrix. (**b**) Comparison of the true (red) and approximate (green) cumulative spectral density of the Laplacian spectrum. (**c**) Minnesota graph with partition obtained by the graph separator.

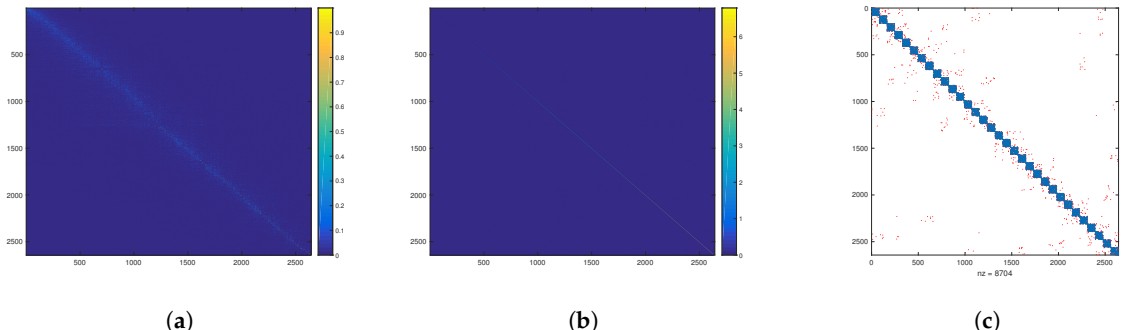

**Figure 7.** (**a**) Absolute value of inner products between the approximate, $\hat{U}$, and the true graph modes, i.e., $|[\hat{U}^T U]_{i,j}|$. (**b**) Absolute value of the approximate diagonalization of the original Laplacian, i.e., $|[\hat{U}^T L \hat{U}]_{i,j}|$. (**c**) Comparison of the original permuted graph (red and blue) and its block diagonal approximation (blue).

## 7. Approximation of Graph Filtering

The main workhorse of graph signal processing is graph filtering. In many instances of the GSP field, we appreciate the *distributable* capabilities of a graph filter. However, this term is only applicable when the communication graph matches the data graph. When all the data resides on a single location, e.g., centralized data management, a distributed graph filter does not necessarily lead to an efficient distributed implementation, i.e., a parallel-friendly computation. As noticed in [5], the main bottleneck of the polynomial graph filter implementation is the queuing effect when performing local computations, i.e., performing a dense matrix-vector multiplication might end up more efficient than sequential sparse matrix-vector multiplications (queuing effect). In [5], the authors propose to compute support-disjoint sparse matrices to perform an approximate GFT. This somehow effectively alleviates the queuing problem as the different matrix-vector multiplications can be carried out in parallel. However, despite that this approach provides a parallel-friendly model for data processing, it still requires a shared memory location, i.e., the matrices operate over the full data vector regardless of the matrix support.

Data most of the time is *geographically restricted.* That is, the transmission and storage of data are usually constrained by the distance to the source. For example, although it is possible to mirror all possible information from different networks in different geographical regions, their storage is often local, i.e., the most accessed data is usually related to the local region (the last use/generated—first fetch principle for caching). Therefore, it makes sense to keep processing local as well and establish a hierarchical

processing chain to minimize both the data management and communication protocol complexity. Luckily, the previously discussed recursive graph partition provides an alternative to deal with data processing in such domains. This is due to the fact that geographically restricted data, usually spawn from social relations or sensor networks, can be (approximately) modeled using planar or geometrical graphs which are structured graphs admitting good recursive graph separators. Examples for such networks can be found in traffic networks or range-limited transmission wireless networks, as discussed in Section 3.

To leverage the recursive decomposition presented in Section 5 for fast approximate graph filtering, we propose to use this hierarchical graph decomposition to compute disjoint graph filters that can be solved locally within the localized node clusters given by the partition. More specifically, we can state this in the following way. Considering a graph $\mathcal{G}$, and a hierarchical decomposition $\mathcal{T}$ [cf. Figure 3]. We approximate the *action* of a desired shift-invariant graph filter [37], $H$, defined by a set of $K$ polynomial coefficients $\{\alpha_k\}_{k=0}^{K}$ and a graph matrix representation $S$, e.g., graph shift-operator in GSP [1] using the structure in $\mathcal{T}$. That is, we approximate $y = Hx$ with

$$\tilde{y} = \mathrm{merge}\left(\{H_t x_t\}_{t \in \mathrm{leafs}(\mathcal{T})}\right) \tag{8}$$

where $H_t$ and $x_t$ are the shift-invariant graph filter and the graph signal partition associated with the leaf $t$ of the partition $\mathcal{T}$, respectively. Here, the simplest merge operation could be just the union of the result of the disjoint node partitions, however, more complex merging routines are possible.

In the following theorem, we provide the first result in approximate graph filters by means of graph bisection for the union merging routine and for a generic matrix representation of the graph, $S$, typically known within GSP as the *graph shift operator*, see, e.g., [2].

**Theorem 5.** *Let $\mathcal{G}$ be a graph, $S$ its matrix representation satisfying $\|S\|_2 < 1$ and $H = \sum_{k=0}^{K} \alpha_k S^k$ a graph filter defined on $\mathcal{G}$ such that $\|Hx\|_2 = \|x\|_2$, where $\mathbf{x}$ is a signal defined over the nodes of $\mathcal{G}$. Then, the block diagonal matrix $\tilde{S}$, obtained by the bisection of $\mathcal{G}$ and removal of the connections between the connected components, can implement a graph filter $\tilde{H} = \sum_{k=0}^{K} \alpha_k \tilde{S}^k$, over the induced graph $\tilde{\mathcal{G}}$ such that*

$$\frac{\|Hx - \tilde{H}x\|_2}{\|Hx\|_2} = \mathcal{O}(\epsilon), \tag{9}$$

*where $K$ is the order of the graph filter and $\epsilon \triangleq \|S - \tilde{S}\|_2$.*

**Proof.** See Appendix F.  □

This result shows that even for graphs with good partitions, there is no straightforward asymptotic efficient approximation for graph filtering. However, it encourages its use for the families presented in this work as the growth rate of the error is not fast. That is, by inspection it can be seen that $\epsilon$ is directly proportional to the graph edit distance, i.e., if $S$ is considered a scaled adjacency matrix then $\epsilon$ can be directly upper bounded with the Frobenius norm which translates to the edit distance. Hence, for the family $\mathcal{S}_{1/2}^l$ the error grows at most following a square-root law in the size of the graph. Numerical experiments in the next section illustrate this behavior.

From the result of Theorem 5, where a single bisection is considered ($d = 1$), we can expect that for higher order recursions, i.e., decomposition depth, the graph filter approximation quality degrades further. The reason for this issue is that the difference between the approximation error of the matrix representation of the graph only worsens with increasing decomposition depth. Hence, a *stitching* method is required in order to improve the approximation at the cost of increasing the communication and implementation cost

of the method. This is left for immediate future research, where the low-rank structure of the off-diagonal blocks, guaranteed by the separator theorem, can be leveraged to obtain a better merging strategy than only working on the leaves of the decomposition.

### 7.1. Results for Approximate Graph Filtering

In the previous discussion, it was shown that the approximation quality for graph filters is $\mathcal{O}(\epsilon)$, where $\epsilon$ is related to the difference between the matrix representation of the graphs. For the graph family $\mathcal{S}_{1/2}^l$, this parameter has an order given by $\mathcal{O}(\sqrt{n})$. Hence, it is expected that the error, for different filter orders, increases following a square-root law. In the following numerical experiments, we illustrate this behavior for approximating a heat kernel filter, i.e., $h(\lambda) = e^{-\tau\lambda/\lambda_{\max}}$, where $\tau = 10$ and $\lambda_{\max}$ is the maximum eigenvalue of the matrix representation of the graph. All the filters evaluated are implemented and designed using the GSP Toolbox [38]. For these experiments, we evaluate the performance of the approximation in terms of the normalized error $e_n = \|Hx - \tilde{H}x\| / \|Hx\|$, where $\tilde{H}$ is the approximate graph filter as described before. Here, we have considered four types of inputs: (i) uniform distributed signals with entries in the range $[0,1]$ (positive/uniform), (ii) zero-mean normalized Gaussian distributed signals (Gaussian), (iii) binary signals (Binary) and (iv) bipolar signals with entries in $\{-1,1\}$ (Bipolar). In Figure 8, the results for these comparisons are shown. From Figure 8a, we can observe the square-root law for the increase on the approximation error as described in the theory. Here, we performed the comparison using a filter order $K = 30$ for both the true and approximate filter. The increasing trend of the error is only reflected in the Gaussian and Bipolar signals as those kinds of signals are zero mean asymptotically. For the Positive/Uniform and Binary signals, the error decays sublinearly as the size of the graph increases. This is due to the fact that the mean term of the signal dominates the expression of the error. In Figure 8b,c, a comparison for different filter orders is shown. Here, it can be seen that the error remains constant across different filter orders. This result is due to the fact that the tested filter is sufficiently smooth for obtaining good approximations with low-order Chebyshev polynomials. The trend observed in Figure 8a is again observed for the different filter orders.

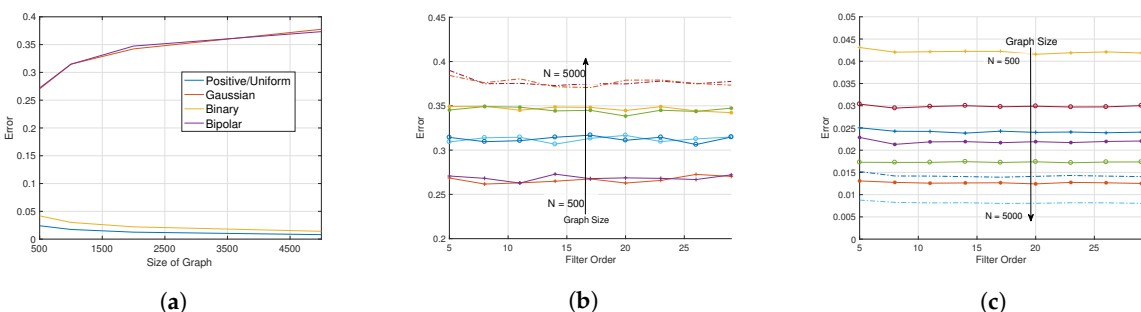

(a)        (b)        (c)

**Figure 8.** Approximation error for graph filtering using the proposed hierarchical decomposition for different kinds of inputs. (**a**) Comparison of the approximation error for graph with varying number of nodes and filter order $K = 30$. Comparison of the approximation error for different filter orders for (**b**) Gaussian and bipolar signals and (**c**) positive and unipolar signals.

### 7.2. A Note on Graph Filter Bank Design

Motivated by the current developments in graph filtering [39–43], there has been a large effort in defining (and designing) filter banks for the task of filtering signals in graphs [17,44,45]. Although there are many alternatives for designing filter banks in general, as remarked in [46], a guiding principle when designing filter banks is to identify gaps in the spectrum of the matrix representation of the graph, e.g., normalized Laplacian or adjacency matrix. For identifying such gaps one requires to have access

either to the full spectrum, which for large-size graphs might not be feasible. Therefore, an approximation method for identifying gaps in the spectrum of such matrices is required.

Here, is where our method provides an approximate solution for designing graph filter banks. In this sense, there are two possible ways to use the proposed method. The first is merely a direct application of our results and estimates the spectrum of the graph by means of the introduced hierarchical decomposition. The second consists of estimating the cumulative spectral density function of the graph [c.f Equation (7)].

For approximating such a function, a widely used method is the *kernel polynomial method* (KPM) [47]. This method has been adapted in [46] using a stochastic estimator of the trace of a matrix which incurs an overall complexity of $\mathcal{O}(KJ|\mathcal{E}|)$, where $K$ and $J$ are the order of the filters used for approximating the Heaviside function and the number of random samples required by the method, respectively. For non-sparse graphs, i.e., $|\mathcal{E}| = \Omega(n)$, this approach might not be appealing as its complexity is not linear anymore in the size of the graph and the number of random samples required by Hutchinson's method for estimating $\mathrm{tr}(A)$, for any given matrix $A$, is known to require $J = 6\epsilon^{-2}\ln(\frac{2}{\delta}\mathrm{rank}\{A\})$ samples for an error of at most $\epsilon$ with probability $\delta$ when Rademacher distributed vectors are employed [48]. Hence, in such cases, the hierarchical decomposition, with deterministic bounds, could be used as an alternative to compute a fast approximation for the cumulative spectral density of the matrix representation of the graph. In this case, using the result of Theorem 2, we can guarantee that our approximation to the cumulative spectral density is asymptotically efficient (for increasing graph sizes) and is related to the edit graph distance. Further, differently from the KPM, our method provides not only an approximation for the CDF but to both eigenvalues and eigenvectors. And if desired, the KPM can be instead employed in the leaves of the decomposition to approximate the CDF at them and then later on construct the global CDF of the graph. This last modification could lead to a significant reduction as the full eigendecomposition of the submatrices at the leaves is no longer required.

## 8. Conclusions

In this paper, we have introduced a set of families of graphs which have good recursive graph separators. Based on this property, we proposed a divide-and-conquer approach for approximating the spectrum of large-sized graphs within these families. The proposed algorithm was stated in general form allowing to generalize the approach to different applications involving the spectrum of a graph. In addition, we derived a theoretical bound on the error for the approximation of the spectrum of such graphs for any depth of the graph decomposition. Furthermore, we showed that the proposed decomposition based on recursive bisections might be beneficial for other graph signal processing tasks where operating with the full graph is infeasible or undesirable. These results form a step forward towards a fast graph Fourier transform analogous to the one existing fast Fourier transform in the time-domain.

**Author Contributions:** Conceptualization, M.C., S.P.C. T.M., G.L.; software, M.C.; formal analysis, M.C.; investigation, M.C.; writing—original draft preparation, M.C.; writing—review and editing, S.P.C., T.M. and G.L.; supervision, S.P.C., T.M. and G.L. All authors have read and agreed to the published version of the manuscript.

**Funding:** This research was funded in part by the ASPIRE project (project 14926 within the STW OTP programme), financed by the Netherlands Organization for Scientific Research (NWO). Mario Coutino was partially funded by CONACYT and AIP Riken.

**Conflicts of Interest:** The authors declare no conflict of interest. In addition, the funders had no role in the design of the study; in the collection, analyses, or interpretation of data; in the writing of the manuscript, or in the decision to publish the results.

## Appendix A. Graph Families with Good Separators

In the following, we present a brief summary of graph families that allow for good separator and that are closed under partitioning, i.e., each element of the partition is within the class.

**Planar Graphs.** A graph that can be drawn without edge crossings is called *planar* [49]. This type of graphs arises typically in applications related to 2-D meshes and computer graphics. For such graphs the following separator theorem is known:

**Theorem A1** ([28]). *Planar graphs satisfy a $\mathcal{O}(n^{1/2})$-separator theorem with constant $\alpha = 2/3$ and $\beta = 2\sqrt{2}$.*

In addition, Lipton and Tarjan also provided a linear-time algorithm to find such a separator in planar graphs. This result is of great importance as for large-scale graphs the computation of such partition remains feasible. Besides planar graphs, degenerate instances of them such as edgeless graphs, linked lists, and trees have good vertex separators [50–52].

**Almost Planar Graphs.** Similar to planar graphs, almost planar graphs can be made planar by solely removing a small number of edges. Instances of this kind of graphs are road networks or power grids. In such (physical) networks, planarity is lost by bridges or tunnels, therefore by removing such edges, the graph can be approximated through a planar one. As a result, the $\mathcal{O}(n^{1/2})$-separator theorem for planar graphs holds (approximately) for this family of graphs.

**Finite Element Graphs.** This family of graphs arises from finite element methods, for example through a tessellation of the space. Following the formal description of Gilbert, et al. a finite element graph can be obtained from a planar graph as follows. First, the graph is embedded in a plane. Next, we identify certain points as *nodes*, e.g., vertices, points on edges, points in faces. Then, edges between all nodes that share a face are *drawn*. From this construction, if the number of nodes per face is bounded by $d$, finite elements graphs satisfy a $\mathcal{O}(dn^{1/2})$-separator theorem [28].

**Graphs Embedded in a Low-Dimensional Space.** Interestingly enough, nearly all graphs that are used to represent connections in low dimensional spaces have small separators. For example, 3-D meshes, under certain nicety conditions, accept a $\mathcal{O}(n^{2/3})$-separator theorem [53]. In general, *unstructured meshes* of dimension $d$ allow a $\mathcal{O}(n^{(d-1)/d})$-separator theorem [53].

**Circuits.** These are one of the families of graphs that motivated the early application of graph separators. This class and its separator are still used for VLSI design when designers want to minimize the area employed [54]. If the circuit, represented through vertices (components) and edges (connections) is drawn with *few* crossings, then it may be considered as an almost planar graph. Hence, the $\mathcal{O}(n^{1/2})$-separator theorem approximately holds. Otherwise, if the circuit can be embedded on a surface of genus [9] $g$, then it has an exact $\mathcal{O}((gn)^{1/2})$-separator theorem [51].

**Geometric Graphs.** This family of graphs arises by construction through *geometrical objects*. As this family of graphs encloses a large variety of objects, e.g., k-nearest-neighbor graphs, meshes, etc., here we only consider the *unit disk* graphs subfamily. Unit disk graphs arise naturally in applications involving sensors networks. These graphs are combinatorial objects generated by the intersection of disks on the plane. They are also known as *Euclidean* graphs. These graphs are constructed from points in the space (vertices) when edges between points are drawn if the distance between them is smaller than some threshold, e.g., range-limit communication graph. For this family of graphs, a $\mathcal{O}(n^{1/2})$-separator theorem holds similarly to the case of planar graphs as they are embeddable in a 2-D surface. For a more in-depth discussion of the results related to this kind of graphs, the reader is referred to [29].

**Social Graphs.** Networks such as *friend*, *bibliographic* or *citation* graphs have good separators in practice as they are based on communities, thus exhibiting local structure, see, e.g, [55]. Within social graphs, most links can be found within some other form of community or local domain, as the link graphs used for the web. Unfortunately, differently from the previously discussed types of graphs, social graphs cannot be guaranteed to accept *good recursive separators* in general. This situation can be observed in a social network represented by a power-law graph. Differently from planar or geometric graphs, despite that a power-law graph can be easily *separated*, there is no guarantee that its partitions accept good separators

themselves as it is not (necessarily) closed under (vertex) edge deletion. However, in practice, this seldom is the case as networks of friends are, again, networks of friends [56].

**Appendix B. Proof of Theorem 1**

Using the results from [57], we can obtain the following inequality for any two graphs $\mathcal{G}$ and $\mathcal{H}$

$$|\mathcal{V}| \cdot W_1(\lambda, \lambda') \leq 2\mathcal{G}\Delta\mathcal{H},$$

where $\mathcal{G}\Delta\mathcal{H}$ denotes the *graph edit distance* [58]. Further, using the fact that $\max_{v \in \mathcal{V}} \deg(v) \in \mathcal{O}(1)$, we have

$$G\Delta\mathcal{H} \in f(n),$$

for some $f$. Therefore, as $|\mathcal{V}| = n$, the result follows.

**Appendix C. Proof of Theorem 2**

First, we prove the the existence of the approximation guarantee. Let us consider the result [57]

$$nW_1(\lambda, \tilde{\lambda}) \leq 2\mathcal{G}\Delta\tilde{\mathcal{G}}, \tag{A1}$$

for the earth mover distance between the spectrum of $\mathcal{G}$ and $\tilde{\mathcal{G}}$. This implies that the approximation error is bounded by the edit distance between the original graph, $\mathcal{G}$, and the graph used to compute the approximate spectrum, $\tilde{\mathcal{G}}$.

The decomposition employed in Algorithm 1, with depth $d$, applied to graphs in $\mathcal{S}_{1/2}$, obtains a surrogate graph $\tilde{\mathcal{G}}$ where

$$
\begin{aligned}
\mathcal{G}\Delta\tilde{\mathcal{G}} &= \alpha n^{1/2} + 2\alpha\sqrt{n/2} + \cdots + 2^{(d-1)}\alpha\sqrt{n/2^{(d-1)}} \\
&= \alpha n^{1/2} \sum_{i=0}^{d-1} \sqrt{2}^i \\
&= \alpha n^{1/2}(1 + \sqrt{2})(2^{\frac{d}{2}} - 1) \\
&= \mathcal{O}(n^{1/2}\sqrt{2}^d - n^{1/2}),
\end{aligned}
\tag{A2}
$$

edges have been removed from the original graph $\mathcal{G}$. Here, we have kept the second term in the big-O notation to provide an expression for zero error when $d = 0$, i.e., if the original graph is considered, the true spectrum must be computed.

A vanishing error for sufficiently large graphs, i.e.,

$$\lim_{n \to \infty} W_1(\lambda, \tilde{\lambda}) \leq \lim_{n \to \infty} \frac{2\mathcal{G}\Delta\tilde{\mathcal{G}}}{n} = 0, \tag{A3}$$

implies that the number of removed edges meets $\mathcal{G}\Delta\tilde{\mathcal{G}} = o(n)$. This ensures that (A3) becomes zero asymptotically. Here, $o(\cdot)$ is used to denote *little-o notation*, i.e., $f(x) = o(g(x))$ implies $\lim_{x \to \infty} f(x)/g(x) = 0$ as $g(x)$ is nonzero, or becomes nonzero after certain point. Therefore, to guarantee this condition we should show that $\mathcal{G}\Delta\tilde{\mathcal{G}} \in \mathcal{O}(n^{(1-\delta)})$ for some $\delta \in \mathbb{R}_{++}$.

If we set the depth of the decomposition to $d^* = \mathcal{O}(\eta_\delta \log n)$ as stated in the theorem, the number of removed edges given by (A2) becomes

$$
\begin{aligned}
\mathcal{G}\Delta\tilde{\mathcal{G}} &= \mathcal{O}(n^{1/2}\sqrt{2}^{d^*} - n^{1/2}) \\
&= \mathcal{O}(n^{1/2} \cdot n^{\eta_\delta} - n^{1/2}) \\
&= \mathcal{O}(n^{1-\delta} - n^{1/2}) \\
&\in \mathcal{O}(n^{1-\delta}),
\end{aligned}
\tag{A4}
$$

as we wanted. Here, we have used the definition $\eta_\delta \triangleq 1 - 2\delta$. Using this result, we can show that the approximation error when employing the proposed decomposition with depth $d^*$ is

$$
\begin{aligned}
W_1(\lambda, \tilde{\lambda}) &\le 2\frac{\mathcal{G}\Delta\tilde{\mathcal{G}}}{n} \\
&= \mathcal{O}(1/n^\delta - n^{1/2}) \tag{A5} \\
&\in \mathcal{O}(1/n^\delta), \tag{A6}
\end{aligned}
$$

where $\delta$ it is seen to define the rate of decay of the error.

From the definition of $d^*$ it is observed that for $0 \le \delta < 1/2$ the depth parameter is positive, which implies a feasible decomposition attaining an $\mathcal{O}(1/n^\delta)$-approximation as desired. For $\delta = 1/2$ we have $d = 0$ which implies that no decomposition is performed and the original graph, $\mathcal{G}$, is used to compute the spectrum. In this case, it is seen from expression (A5) that the error is zero.

The time complexity can be obtained by considering the dominant (competing) operations: (i) computation of eigendecomposition of the leaves, and (ii) computation of the graph decomposition. The complexity for (i), $T_e(n, d)$, is obtained by considering that there are $2^d$ leaves in the decomposition and that each leaf contains a $n/2^d \times n/2^d$ matrix, i.e.,

$$
T_e(n, d) = \mathcal{O}(2^d T_f(n/2^d)),
\tag{A7}
$$

where $T_f(m)$ is the time required by $f(\cdot)$ to compute the eigendecomposition of an $m \times m$ matrix.

The complexity for $(ii)$, $T_d(n, d)$, is obtained considering that there are $d$ levels in the tree, and that at the $i$th level $2^i$ decompositions on graphs with $n/2^i$ nodes are performed, i.e., $\mathcal{O}(d \cdot 2^i \cdot T_g(n/2^i))$. As for graphs in $\mathcal{S}_{1/2}$ the decomposition requires linear time, i.e., $T_g(m) = \mathcal{O}(m)$, we obtain linear complexity in $n$ and $d$, i.e.,

$$
T_d(n, d) = \mathcal{O}(nd).
\tag{A8}
$$

As a result, the total complexity can be expressed as

$$
\begin{aligned}
T(n, d) &= T_e(n, d) + T_d(n, d) \\
&= \mathcal{O}(2^d T_f(\tfrac{n}{2^d})) + \mathcal{O}(nd).
\end{aligned}
\tag{A9}
$$

From (A9), it is observed that the time for computing the eigendecomposition diminishes for larger $d$, while the time for computing the graph decomposition increases with $d$, hence their competing nature.

As most of the (if not all) algorithms for general matrix eigendecomposition have cubic time complexity, $T_f(m) = \mathcal{O}(m^3)$, we obtain the following total complexity.

$$
T(n, d) = \mathcal{O}(n^3/2^{2d}) + \mathcal{O}(nd).
\tag{A10}
$$

Now, substituting the optimal depth parameter, $d^*$, in (A10) we obtain a total complexity

$$T(n, \delta) = \mathcal{O}(n^{1+4\delta} + (1 - 2\delta)n \log n), \tag{A11}$$

depending on the size of the graph and the accuracy of the approximation. Furthermore, considering that

$$\log n = o(n^\omega), \quad \forall\, \omega > 0, \tag{A12}$$

we obtain

$$T(n, \delta) = \mathcal{O}(n^{1+4\delta}), \tag{A13}$$

as we wanted to show.

## Appendix D. Proof Theorem 3

Direct use of the Bauer-Fike theorem leads to the inequality

$$\frac{\|\hat{u} - u\|_2}{\|\hat{u}\|_2} \leq \|(S - \mu I)^\dagger\|_2 (\|E\|_2 + |\hat{\mu} - \mu|), \tag{A14}$$

where we have defined $E := S - \hat{S}$.

Using the fact that $\hat{S}$ is a normal matrix, i.e., unitary eigenvectors, and $|\hat{\mu} - \mu| \leq \|E\|_2$ by the Weyl's inequality [59], we obtain

$$\|\hat{u} - u\|_2^2 \leq \frac{4}{\min_{l \in \text{eig}(S), l \neq \mu} |l - \mu|^2} \|E\|_F^2. \tag{A15}$$

Here, we used the fact that $\|E\|_2 \leq \|E\|_F$ and the definition of the spectral norm of $(S - \mu I)^\dagger$. Thus, the result follows.

## Appendix E. Proof of Theorem 4

Using results from the convergence analysis of the power method, we directly obtain

$$\|u - \hat{u}_{k+1}\|_2 \in \mathcal{O}\left(\left|\frac{\hat{\mu} - \mu}{\hat{\mu} - \gamma}\right|^k\right), \tag{A16}$$

where $\gamma$ is the second closest eigenvalue of $S$ to $\hat{\mu}$. Considering that $|\hat{\mu} - \mu| \in \mathcal{O}(1/n^\delta)$ and making the approximation

$$|\mu + \mathcal{O}(1/n^\delta) - \gamma| \approx |\mu - \gamma|, \tag{A17}$$

for small $\mathcal{O}(1/n^\delta)$, we obtain

$$\|u - \hat{u}_{k+1}\|_2 \in \mathcal{O}\left(\left|\frac{1/n^\delta}{\mu - \gamma}\right|^k\right). \tag{A18}$$

Further, noting that $|\mu - \mu_{\text{close}}| \leq |\mu - \gamma|$, the convergence rate result follows.

To show the result for the computationally complexity, first we recall the following result

**Theorem A2** ([12])**.** *On input an $n \times n$ symmetric diagonally dominant matrix $X$ with $m$ non-zero entries and a vector $b$, a vector $\bar{x}$ satisfying $\|\bar{x} - X^\dagger b\|_X < \eta \|X^\dagger b\|_X$, can be computed in expected time $\tilde{\mathcal{O}}(m \log^2 n \log 1/\eta)$.*

For the case that $S$ is taken as the graph adjacency matrix, if the eigenpairs meet the condition stated in the theorem, we have that $S$ is diagonally dominant. Thus, the complexity result follows.

For the case that $S$ is the combinatorial Laplacian, $L$, we cannot directly use the result of [12] because $S - \hat{\mu}I$ is not diagonally dominant. Hence, we first need to show how to implement the solution of the involved linear system through Laplacian systems. Without loss of generality, we assume that $\mathbf{1}^T\hat{u} = 0$. So, an equivalent linear system can be built as

$$(I - \hat{\mu}L^\dagger)y_{k+1} = \frac{1}{c_k}\hat{u}_k, \tag{A19}$$

where $y_{k+1} := L\hat{u}_{k+1}$. The above linear system can be solved by the Jacobi method, i.e.,

$$y_{k+1}^{(i+1)} = \frac{1}{c_k}\hat{u}_k + \mu z^{(i)}, \tag{A20}$$

where $Lz^{(i)} = y_{k+1}^{(i)}$. Notice that each solve, for both $z^{(i)}$ and $\hat{u}_{k+1}$ is based on a Laplacian system. Hence, the result of [12] can be used and the complexity result follows.

**Appendix F. Proof Theorem 5**

Let us first establish the following.

$$\frac{\|Hx - \tilde{H}x\|_2}{\|Hx\|_2} \leq \frac{\|H - \tilde{H}\|_2\|x\|_2}{\|x\|_2} = \|H - \tilde{H}\|_2. \tag{A21}$$

Therefore, to show the result of the theorem we only need to bound the spectral norm of the difference between $H$ and $\tilde{H}$. This can be done by the following chains of inequalities.

$$\begin{aligned}
\|H - \tilde{H}\|_2 &= \|\sum_{k=0}^{K}\alpha_k S^k - \sum_{k=0}^{K}\alpha_k \tilde{S}^k\|_2 \\
&= \|\sum_{k=0}^{K}\alpha_k(S^k - \tilde{S}^k)\|_2 \\
&\leq \alpha_{\max}\sum_{k=0}^{K}\|S^k - \tilde{S}^k\|_2 \\
&\leq \alpha_{\max}\sum_{k=0}^{K}k\epsilon\|\tilde{S}\|_2^{k-1} + \mathcal{O}(K\epsilon^2).
\end{aligned} \tag{A22}$$

Here, we have considered $\epsilon = \|S - \tilde{S}\|_2$ as stated in the theorem. Now, recalling that $\|\tilde{S}\|_2 < 1$ and $\epsilon \leq 1$ by definition, we can further bound (A22) as

$$\begin{aligned}
\|H - \tilde{H}\|_2 &\leq \alpha_{\max}\sum_{k=0}^{K}k\epsilon\|\tilde{S}\|_2^{k-1} + \mathcal{O}(K\epsilon^2) \\
&\leq \alpha_{\max}\sum_{k=0}^{\infty}k\epsilon\|\tilde{S}\|_2^{k-1} \\
&= \frac{\alpha_{\max}}{(1 - \|\tilde{S}\|)^2}\epsilon,
\end{aligned} \tag{A23}$$

where the identity

$$\sum_{k=0}^{\infty}kx^{k-1} = \frac{1}{(1-x)^2} \quad \text{for } |x| < 1,$$

has been used in the third inequality and the second-order error term has been dropped.

Finally, as it is assumed that the spectral norm is strictly upper bounded by one, i.e., $\|\tilde{\boldsymbol{S}}\| \in \mathcal{O}(1)$, we obtain the desired result.

$$\|\boldsymbol{H} - \tilde{\boldsymbol{H}}\|_2 \in \mathcal{O}(\epsilon). \tag{A24}$$

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
