# Peer review of "Fast Spectral Approximation of Structured Graphs with Applications to Graph Filtering"

_algorithms, doi:10.3390/a13090214_

Round 1

Reviewer 1 Report

See attached file.

Reviewer 2 Report

This is a very good paper. It is well written, well-structured and easy to read. The statement of the problem is clearly defined and the precedents are mostly complete. The paper provides new significant contributions to a hard topic: finding computationally efficient algorithms to calculate the GFT. Unfortunately there is not possible a powerful and general algorithm like FFT but the authors convincingly demonstrate that for some families of graphs and under reasonable approximations, very significant computational saving is possible by a so called divide-and-conquer approach.

Just one small correction: page 6, the numbering of the first equation is missing.

And one suggestion. FFT can be directly applied to real and complex signals. Different forms of Complex Graph Fourier transform (which have GFT as a particular case) have been proposed in 1, 2 and 3 (below). It would be most interesting to comment on the possible generalization of the algorithm to the complex case, even though it may be the subject of future research.

  1. Yu, G.; Qu, H. Hermitian Laplacian Matrix and positive of mixed graphs. Applied Mathematics and Computation, 2015, 269, 70-76.
  2. Wilson, R.C.; Hancock, E.R. Spectral Analysis of Complex Laplacian Matrices. In Structural, Syntactic and Statistical Pattern Recognition LNCS, Fred, A., Caelli, T.M., Duin, R.P.W., Campilho, A., de Ridder, D., Eds. Springer: Heidelberg, 2004, 57–65.
  3. Belda, J., Vergara, L., Salazar, A., Safont, & G. Parcheta, Z. (2019). A new surrogating method by the Complex Graph Fourier Transform. Entropy, 21-759 (2019), 1-18.

Reviewer 3 Report

The manuscript presents a method to compute approximations of the spectrum of a graph operator (adjacency matrix or Laplacian) in a fast manner on particular families of graphs by using a divide-and-conquer approach. This is made possible thanks to the properties of these families. The results presented are original and interesting. I would recommend the publication of the manuscript after minor revision. Here are my remarks.

General remarks.
* The authors claim in the introduction to both provide an estimation of the eigenvalues and the eigenvectors. However, the bounds and experiments are focused on the eigenvalues. There are no result about the eigenvectors or I may have missed them?
* It is not clear to me how the authors perform the graph bisection. I understand that they refer to ref. [23]. The paper would be clearer if they could give more details on the bisection approach and its complexity, both in the theory and in their experiments. The search for a good (and balanced) recursive bisection might be more time consuming than the computation of eigenvalues of the subgraphs, in particular for very large graphs. It would be interesting to have more details for the reproducibility of the experiments, for example how was performed the bisection on the Minnesota graph?
* The merging of the eigenvalues could be discussed more. Only one sentence describes the merging process in Sec. 5.1, and this merging is only a sorting of the eigenvalues. Could they discuss the other possibilities if any?
* My hypothesis is that the smallest eigenvalues are the most perturbed by the bisection of the graph as they correspond to smooth eigenvectors (with small variations) and non-zero values all over the graph (at least for the Laplacian case). Large eigenvalues often correspond to localized eigenvectors (with zero values on a large part of the graph), less impacted by a cut of the graph. The perturbation of the smallest eigenvalues may stay reasonably small, as their value is small compared to the values of the largest eigenvalues. I would be curious to know if this is true if the authors have time to make an additional experiment that shows the error on the smallest eigenvalues and on the largest ones. This is related to Section 7.1 where there is an experiment with a filter that is exponentially decreasing on the spectrum. The filter will be affected more if the smallest eigenvalues are more perturbed than the largest ones. The authors could test a second filter which is zero for small spectral values and non-zero on the largest eigenvalues.

Abstract
Talking about the butterfly pattern in the abstract may be a bit misleading. The reader may think a similar pattern could be derived on graphs, together with an exact FFT (not approximated). This is not the case as the authors present an approximated version of the Fourier transform and it is not possible as this fast transform requires a very strict condition: the number of samples must be a power of 2. So even in the regular path graph, the Fast Fourier transform can not be performed if the number of nodes is not appropriate. In addition, the authors state later on in the motivation that they want to compute the Fourier modes rather than the Fourier transform and the butterfly pattern is used for the Fourier transform, not for computing the modes (which are already known).

Introduction
A sentence stating what are the graph modes would be welcome (eigenvectors of the graph Laplacian or another operator?). Maybe with a reference to the definition of Fourier modes such as [Shuman et al, vertex-frequency analysis on graphs, ACHA (2016)] or to the recent review [ricaud et al, Fourier could be a data scientist: From graph Fourier transform to signal processing on graphs, Comptes Rendus Physique (2019)]

"eigenfunctions of the continuous manifold" does not make sense, it must be eigenfunctions of an operator over the continuous manifold. And which operator? The authors suggest the adjacency matrix for a graph but what is the equivalent for a continuous manifold? I would suggest using the Laplacian instead which have a clear definition on any (smooth) manifold.

Section 3.2. I would replace the foot note 1 to a reference to Appendix A directly in the text or even motivate some examples of graphs inside the text. The examples given in Appendix A are very important to understand the families of graph that satisfy definition 1 and to understand the manuscript. If possible with a reference to Figure 1 and some explanation of what would be for these graphs A, B and C.

Ref. [44] has been published in ACM, please update the reference. It would be good to add more details in the proof of Theorem 1, to see where the properties of G to have (alpha,beta)-separator are useful. Result from [44] is for any graph G and H (would be good to remind the conditions in the appendix).

Section 6

Could the authors remind what is the matrix they want to approximate the eigenvalues from? Is it the adjacency matrix?

In section 6.1. The authors generate 100 random graphs within the S_{1/2} family. "The model to generate such graphs follows a unit disk model". They should refer to Appendix A where it is defined. Could they provide more details on how they generate these graphs? What is the space dimension, what is random, how it is randomly distributed?

Section 6.2. In this section, the number of bisections is independent of the size of the graph. Hence, a larger graph is decomposed in larger subgraphs. The error decrease shown in figure 5 could be due to the fact that the subgraphs are larger and their eigenvalues better approximate to the ones of the initial graph. If I understand well with d=6, the initial graph is cut into 2^6 = 64 pieces, so that for n=1000, the subgraphs contain between 10 to 20 nodes, which is rather small. Second question: What happens if the subgraphs are still too large and have to be further bisected? In other words, how does the error evolve with increasing initial graph size but keeping constant subgraphs size (adapting the depth to get subgraphs of close size)?

Section 7
In theorem 3, the operator S tilde and the spaces where the operators act are not defined. "S tilde obtained by a bisection", I am not sure what it means: is it a block diagonal matrix where each block is the operator S restricted to a subgraph? The L2 norm is for the Hilbert space of vectors taking values on the nodes of the initial graph? Please make it more explicit.

In appendix A, the "miscellaneous graphs" part should be rephrased. Moreover, it is claimed that these graphs have good separators and later in the paragraph "power-law graph can be easily separated". Could the authors provide some evidence of this, some reference or develop why they think it is the case?

In appendix D, in (A15) How does the last line is obtained from the one above? It is not clear to me. Also, on the last line, it would be better to start the sum at k=1.
in (A16) it should be inequalities instead of equalities.

In the conclusion:
" These results form a step forward towards a true approximate fast graph Fourier transform"
What does a "true approximate" means?

Round 2

Reviewer 1 Report

The authors have generally addressed my comments in this revision. I recommend the paper be accepted, but do have a few minor suggestions and clarifications to my original comments:

  1. There is still no mention of the edge weights. I understand the normalized Laplacian is the only operator of those considered whose spectral range is not drastically affected by the inclusion of large weights. However, it would be helpful to explicitly mention that the normalized Laplacian considered here can be based on a weighted, undirected graph and mention that A is not necessarily a 0/1 matrix.
  2. Does Theorem 3 assume that the eigenvalues are unique?
  3. I do think it would be helpful to include something along the lines of Theorem 2 in the reviewers response. It might be even more helpful to the reader to include an example of how the initial (localized) estimation of a smooth (nonlocalized) eigenvector can be improved by additional processing, since these eigenvectors are used in so many different applications.
  4. It is still not completely clear to me when this method should be preferred to the kernel polynomial method and Lanczos method of [46] when approximating the spectral density function. The authors mention non-sparse graphs, and that is clear. For sparse graphs, this method seems slower, but possibly more accurate. Is that correct? The estimation of the eigenvectors as well may be an additional benefit that the authors might highlight a bit more.
